# Explore, Design and Act for Sustainability: A Participatory Planning Approach for Local Energy Sustainability

**Elaine Fouché [1],\* and Alan Brent [2]**

[1]  Department of Industrial Engineering and the Centre for Renewable and Sustainable Energy Studies, Stellenbosch University, Stellenbosch 7602, South Africa
[2]  Department of Industrial Engineering, Stellenbosch University, and Sustainable Energy Systems, School of Engineering and Computer Science, Victoria University of Wellington, Wellington 6140, New Zealand; alan.brent@vuw.ac.nz
\*  Correspondence: foucheelaine@gmail.com

**Abstract:** This paper focuses on the development of a participatory planning approach for local energy sustainability. The characteristics of a complex problem were reviewed to establish that the problem of sustainable energy at a local government level is complex. In order to better manage complex problems, the literature shows that soft operational research or problem-structuring methods need to be applied, and hence these methods were used as a starting point for developing a participatory planning approach. The requirements for a planning approach were elicited, namely that the approach must be participative and inclusive, holistic, simple and transparent. In addition, the approach must include the identification and assessment of risks as part of the deliberation process, the development of a realistic action plan must be attainable at the end of the stakeholder engagement, the approach must be dynamic, and should be formalised with clear institutional arrangements. A novel participatory approach, namely EDAS—to Explore, Design and Act for Sustainability—was then developed, applied, and evaluated as part of a case study with a local municipality in the Western Cape Province of South Africa. The insights are relevant not only for local governments, but for any institution on a journey towards sustainability.

**Keywords:** energy sustainability; local government; problem structuring; stakeholder involvement; climate change; group decision support; action research; collaborative governance; deliberation

## 1. Introduction

Moving to a low-carbon energy economy requires changing the current energy landscape. This is a complex problem encompassing a broad set of aspects, such as changes in technologies, energy networks and infrastructure, social practices, public attitudes, policies and regulations, to name a few [1–4]. Consistently providing affordable energy services, achieving security of energy supplies and reducing carbon emissions require the deployment of low-carbon technologies and energy-efficiency measures, of which the costs and benefits are often uncertain [2].

Sustainable energy is a principle in which the use of energy "meets the needs of the present without compromising the ability of future generations to meet their own needs", and has two key components, namely: renewable energy, and energy efficiency [5] (p. 10). Renewable energy is generated by resources that are self-replenished, such as wind, solar, biomass and hydro power [6]. Energy efficiency, on the other hand, includes using less energy (kWh) to achieve the same benefits [5].

In order to deal with the uncertainties and complexities in the transitioning to a sustainable energy system, it is argued that diverse stakeholders must be involved from the start of the process [7–9].

Although many problem-structuring methods (PSMs) [7,10] and other participatory approaches [11] are available in literature, these methods have not been specifically developed to plan for energy sustainability for a local government context in South Africa. Public participation is a democratic right of all South African citizens, as stipulated in the Constitution of South Africa [12], but limited evidence could be found in the literature on how public participation is being facilitated, as well as on its effectiveness. The literature further shows that the engagement of public agencies and non-state holders in collective decision making or collaborative governance is still in its infancy in South Africa [13,14]. The aim of this paper is therefore to close the research gap in proposing a participatory planning approach to plan for local energy sustainability in a developing country such as South Africa.

Section 2 presents the materials and methods used to conduct the research. A conceptual contribution is made through the development of a novel participatory approach, namely Explore, Design and Act for Sustainability (in short, EDAS). The application and evaluation of the EDAS approach in a local government context in South Africa further provide a practical contribution to the literature that can be used as an example when similar interventions are planned. A further contribution is the provision of a practical participatory planning approach as a case study for collaborative governance in the field of public administration. This paper is organised into six sections. Section 1 provides the introduction. Section 3 gives the results from the literature review and presents the characteristics of a complex problem, the problem of local energy sustainability, an overview of current problem-structuring and decision-making methods in the field of soft operational research (OR) and an overview of other relevant literature, namely risk assessment methods, public participation, and collaborative governance. The requirements of a participatory planning approach for local energy sustainability, which have been used as the basis in the development of EDAS, are then discussed. Section 4 introduces the conceptualisation of the EDAS approach and discusses how the approach was applied and evaluated in a participatory workshop with Hessequa Municipality, a local government in the Western Cape Province, South Africa. Section 5 offers a discussion of the research results, the implications of the research results, the lessons learnt and future research opportunities. Finally, Section 6 discusses the limitations of the study and makes concluding remarks.

## 2. Materials and Methods

In order to develop a participatory planning approach for local energy sustainability, a participatory action research approach was used in a case study of Hessequa Municipality in the Western Cape Province of South Africa. The action research was conducted over a period of five years, which allowed time for the main researcher to build an understanding of the local government context and to form a trust relationship with the municipal management and stakeholders. The action research methodology provided a cycled approach that allows for collaboration and reflection throughout the different cycles.

For the research on which this paper focuses, a literature review was conducted to conceptualise the EDAS approach. Firstly, a review of the characteristics of a complex problem was conducted to evaluate the sustainable energy decisions at a local government level against these characteristics. Secondly, a review of the current problem-structuring methods (PSMs) in the field of soft OR, risk assessment methods and the literature on public participation and collaborative governance informed the conceptual development of the new approach. The decision to focus on soft OR methods was based on the limited data available during the planning phase of a project, multiple stakeholders, all with different perspectives, and the complex nature of sustainable energy. The requirements of a participatory planning approach were elicited using inductive reasoning and through working closely with local government over a five-year period.

Finally, to evaluate the developed participatory planning approach, a facilitated workshop was conducted at Hessequa Municipality, a local municipality in the Western Cape Province of South Africa. The nature of the workshop was qualitative, using both divergent and convergent collective thinking. Data collection took place in the form of voice-recorded open discussions, group discussions and group feedback. The workshop was concluded with the completion of an evaluation form to validate the

approach that was followed. The data of the workshop were analysed and reported to the municipal management team for sign-off.

## 3. Results from a Literature Review

### 3.1. The Characteristics of a Complex Problem and the Problem of Local Energy Sustainability

In order to determine the characteristics of complex problems, the definitions from the literature, in Table 1, were used. The first characteristic that clearly stands out from the definitions is that a complex problem consists of multiple stakeholders with different views and perceptions of the problem [15]. When there are different views, there will always be multiple objectives, where different stakeholders have diverse opinions on what should be achieved as part of decision making. In addition, a complex problem lacks structure and is seen as a system of problems with many interrelationships. Incomplete, contradictory and changing requirements [16] show us that a complex problem is characterised by uncertainty and risk. Uncertainty and risk when implementing possible solutions to better manage a wicked problem could be high because of unintended consequences, which are very difficult to recognise before implementation [17]. According to Pidd [15], the main difference between uncertainty and risk is that uncertainty cannot be measured, whereas risk can be measured because the probabilities of certain outcomes are known or attainable. Marczyk [18], on the other hand, argues that risk rating is a redundant, intangible concept, because probability is not something that exists in nature.

As determined, complex problems require holistic approaches to plan and manage them. Marczyk [18] argues that a complexity-based, holistic approach, focusing on short-term actions, is needed to manage risks. It is clear from the 10 characteristics of a wicked problem [17] that every complex or wicked problem is unique and that there is no best practice method to address such a problem. Furthermore, complex problems cannot be solved, because every wicked problem is a symptom of another problem. Cilliers [19] argues that complex systems can be influenced, but not controlled, and that no single model can capture all the properties of a complex system. From these characteristics of complex problems and their non-linear properties it can be concluded that reductionism [20], which aims at analysing and finding definite solutions, cannot be used when dealing with complex problems.

**Table 1.** Definitions of the characteristics of a complex problem.

| Characteristic of a Complex Problem | Definitions and Statements Found in Literature |
|---|---|
| Multiple stakeholders/stakeholder perspectives and/or multiple objectives | "A mess is a system of external conditions that produces dissatisfaction" [20] (p. 5), meaning a set of circumstances in which there is extreme uncertainty and in which there may well be disagreement. "A mess is a system of problems with multiple stakeholders who may quite hold different views of what is feasible and desirable" [15] (p. 46). "There is no definitive formulation of a wicked problem" and "The information needed to understand the problem depends upon one's idea for solving it" [17] (p. 161). "[P]roblems are constructs of the human mind and of people working together" [15] (p. 55). "The nature of the decision makers will also greatly affect the type of solution needed" and "The major factor of interest here concerns the objectives of the decision makers" [21] (p. 476). Jackson and Keys classify complex problems as pluralistic, due to stakeholders having divergent views about goals and objectives [21]. |
| Interrelated issues/variables/ factors and a complex structure/system | "In a mess, there are many issues to be faced, they are interrelated, and the interrelationships are often as important as the issues themselves" [15] (p. 46). An ill structured problem "is usually defined as a problem whose structure lacks definition in some respect" [22] (p. 181). Complex problems cannot be addressed in a piecemeal way or solved in full. Complex problems have to be engaged with directly and result from networks of multiple interacting and emerging causes that cannot be individually distinguished [23]. Every wicked problem is a symptom of another problem [17]. "Managers are not confronted with problems that are independent of each other, but with dynamic situations that consist of complex systems of changing problems that interact with each other" [24] (p. 99). |

**Table 1.** *Cont.*

| Characteristic of a Complex Problem | Definitions and Statements Found in Literature |
|---|---|
| Uncertainty and risks | "[T]he term 'wicked problem' refers to that class of social system problems which are ill-formulated, where the in-formation is confusing, where there are many clients and decision makers with conflicting values, and where the ramifications in the whole system are thoroughly confusing" [16] (p. 141). Wicked problems, when implemented, "will generate waves of consequences over an extended–virtually an unbounded–period of time" [17] (p. 163). "Every solution to a wicked problem is a 'one-shot operation'; because there is no opportunity to learn by trial-and-error, every attempt counts significantly" [17] (p. 163). |

The decision to pursue sustainable energy projects at a local government level in South Africa is complex. The direct stakeholders of a municipal area are the local citizens, business owners, tourists and visitors, the municipal administration team, as well as the municipal council. In the energy context, other stakeholders are the farmers in the area (who receive their electricity from Eskom, the national state-owned power utility, directly), the district municipality, local government and other regulatory bodies, such as the South African Local Government Association, the National Energy Regulator of South Africa, the Treasury, the Department of Energy and the Department of Environmental Affairs. Each of these stakeholders will have different viewpoints and end goals when discussing the sustainable energy future of a municipality. Previous research conducted by Fouché and Brent [25] showed the interrelatedness of renewable energy in terms of many aspects of the municipality, such as environmental matters, the municipal infrastructure, the financial system of the municipality, future growth and development, social cohesion, people development and the services provided by either local government or district government. Implementing a change in one part of the system can have ripple effects and unintended consequences in other parts of the system. It is therefore important to consider these unintended consequences when planning for sustainable energy.

The energy, economic and political landscapes in South Africa are faced with many risks and uncertainties. Eskom, which produces and supplies 95% of all electricity in South Africa, is facing problems with increasing debt levels, unstable supply of electricity, resulting in ad hoc load shedding country-wide, and labour unrest [26–28]. A stagnant economy [29], coupled with political instability [30], contributes to a risky and uncertain environment. Therefore, in general, pursuing sustainable energy projects at a local government level is complex and entails many stakeholders, diverse opinions, multiple objectives, a complex structure, and many risks and uncertainties. In order to ensure the proper planning and management of this complex problem, a holistic approach is needed for application in a local government context in South Africa.

### 3.2. Current Problem-Structuring and Decision-Making Methods in the Field of Soft Operational Research

Soft OR stems from Ackoff's realisation [31] that traditional OR methods, focusing on finding optimal solutions for a given problem, do not necessarily focus on the right or total problem. These traditional OR methods focused on quantitative data only, omitting qualitative data such as the viewpoints and perceptions of different stakeholders. In addition, traditional OR methods do not adequately deal with uncertainty [32]. Since 1961, many academics in the OR community [33–36] have echoed Ackoff's concerns to find ways to determine what the real problem is before trying to solve the problem [37]. The need for approaches that consider different perceptions from multiple stakeholders led to the development of soft OR methodologies. Rosenhead [38] identified that these soft OR methodologies constituted a new paradigm of analysis compared to traditional OR and, since then, the consistent naming convention of 'problem-structuring methods' emerged. Rosenhead and Mingers [32] further established the characteristics of these PSMs, namely that they are non-optimising, are less dependent on quantitative data, integrate hard and soft data with social judgement, are simple and transparent, conceptualise people as active subjects, facilitate planning from the bottom up, accept uncertainty and aim to keep options open. Recently, Smith and Shaw [39] highlighted through an exploratory review of the literature that four additional characteristics of PSMs can be added based

on a four-pillar framework that focuses on system characteristics, knowledge and the involvement of stakeholders, the values of model building and structured analysis. These additional characteristics of PSMs are as follows: the approach identifies a system to model, the model-building process is generic and transferrable to multiple problem contexts, the approach structures knowledge through different stages of analysis and the approach has distinct phases of divergent and convergent thinking. The major PSMs that confirm all the characteristics, as listed by Smith and Shaw [39], are soft systems methodology (SSM) [40], the strategic choice approach (SCA) [41], strategic options development and analysis [42] and robustness analysis (RA) [32].

### 3.3. Risk Assessment Methods

As discussed previously, the energy, economic and political landscapes in South Africa are faced with many risks and uncertainties. In addition, sustainable energy projects or energy innovation projects are characterised by many risks [43], such as numerous risks from the environment, which are hard to predict, multidisciplinary project teams and multiple stakeholders, difficult decision making and financial risk. In order to plan for a sustainable energy future at a local government level, these risks and uncertainties need to be considered, but according to Bowers and Khorakian [44], approaches to deal with these risks have not been explicitly examined. A variety of risk assessment techniques from multiple disciplines are summarised by the American Society for Safety Engineers and given in Luko [45]. In the early stages of project planning, qualitative methods such as brainstorming, interviews, Delphi analysis, checklists, structured 'what-if' analysis and the consequence/probability matrix are deemed suitable [44]. These techniques do not require intensive data demands and are dependent on the subjective opinions of the stakeholders. More recently, Marczyk [18] took a radically new approach in defining risk and developing a complexity-based approach to dealing with uncertainty and risk management. The fundamental components of Marczyk's approach are a complexity map, a complexity-based rating of the organisation as well as its ecosystem, a stability index and a complexity profile.

### 3.4. Literature on Public Participation and Collaborative Governance

In a democracy such as South Africa, community participation is not new and forms part of South African and local government policies [46]. One of the terms used for involving citizens in local governmental matters is 'public participation', which is a democratic process that provides individuals and groups from the community with an opportunity to influence socio-political and economic conditions for the better [47]. Public participation involves two-way communication, negotiation and the development of mutual understanding, with the ultimate objective of reaching decisions that are supported by the public. During the process of public participation, citizens' concerns, needs and values are incorporated into governmental decision making [48]. Another well-cited approach found in the field of public administration and policy making is collaborative governance, a form of governance to replace adversarial and managerial modes of policy making and implementation [49]. Collaborative governance is defined as "a governing arrangement where one or more public agencies directly engage non-state stakeholders in a collective decision-making process that is formal, consensus-oriented, and deliberative and that aims to make or implement public policy or manage public programs or assets" [49] (p. 544). Ansell and Gash pointed out a series of factors for successful collaboration, namely face-to-face dialogue, trust building, the development of commitment and shared understanding [49]. Kamara argued that collective governance is still in its infancy in South Africa [13]. Leck and Simon [14] showed that maximum effectiveness of local government initiatives can only be achieved when collaborative governance includes all complementary skills and resources from all relevant private, local, regional and national institutions. Gollagher and Hartz-Karp [50] offered deliberative collaborative governance as a logical hybrid of two closely related terms, namely 'deliberative democracy' and 'collaborative governance', for dealing especially with complex problems such as sustainability issues. Deliberative collaborative governance aims to include the public in its full

demographic diversity, ensuring deliberation (weighing of policy options and consequences) and exerting influence. The main rationale for both collaborative governance and deliberative collaborative governance is to inform government decision making; to alter existing government decision-making outcomes, processes or structures; and to open the possibility of collaborative governance beyond or without government [50]. Although deliberative collaborative governance is a novel idea, the authors felt comfortable, in a South African context, to use public participation and collaborative governance as the overarching frameworks in the development of the EDAS approach.

*3.5. The Requirements of a Participatory Planning Approach for Local Energy Sustainability*

When selecting or developing an approach to plan for a sustainable energy future at a local government level, both the literature and the local context should be considered to define the requirements for a planning approach. The requirements for such a planning approach, as elicited from the literature and confirmed with local government management, are as follows:

3.5.1. The Approach Must Be Participative and Inclusive

In order to include the perceptions and viewpoints (beliefs, interests, values and worldviews) of multiple stakeholders, the approach needs to be participative. Mingers and Rosenhead [10] emphasised that a PSM has to enable several alternative perspectives to be brought into conjunction with one another, something that can only be achieved when all the stakeholders are given an opportunity to share ideas. In addition, the approach needs to be able to make the vast amount of information more accessible, and the problem must be structured in such a way that the richness of information across multiple problem dimensions is not lost [51]. The information shared should be cognitively accessible to actors with different backgrounds and skill levels [10].

3.5.2. The Approach Must Be Holistic

As already discussed, the non-linear properties of complex problems can only be dealt with using a holistic approach, such as systems thinking [40,52]. Maani and Cavana [52] defined systems thinking as a scientific field of knowledge for understanding change and complexity through the study of dynamic cause and effect over time. Checkland [40] used the notion of systems thinking in SSM to represent the real world in a conceptual model, which shows interconnected human and organisational factors in the way that they are perceived by stakeholders.

3.5.3. The Approach Must Be Simple and Transparent

The requirement for simplicity and transparency is based on Miller's observations [53] that the human mind can only retain five to seven units of concentration at any given time, as well as Simon's theory of bounded rationality [54]. The term 'bounded rationality' considers the cognitive limitations of the decision-maker in terms of both knowledge and computational capacity. Bounded rationality is concerned with the ways in which the actual decision-making process influences the decisions that are reached. Simon [54] argues that the process of decision making should seek for satisfying decisions, rather than aiming for the most optimal decisions. Transparency can be attained in both the visual representation and capturing of an audit trail of the facilitated process. Rich pictures can be used to structure problems in order to help groups and individuals to understand the complex context of a situation [40]. Rosenhead's RA [32] and Friend's SCA [41] used a diagrammatic representation of the different strategies discussed during the facilitated process. The advantages and disadvantages of visualisation techniques, such as visual metaphors, conceptual diagrams, mind maps [55] and concept maps [56], were summarised in Eppler [57]. The advantages of these visualisation techniques for use during a facilitation session include rapid information provision, providing a concise overview of the complex situation, emphasising relationships and connections between concepts, encouraging creativity and self-expression, drawing attention and inspiring curiosity.

3.5.4. The Approach Must Include the Identification and Assessment of Risks as Part of the Deliberation Process

In order to ensure that all viewpoints and potential options are considered during the participatory approach, divergent thinking followed by deliberation should be an important aspect. Deliberation is the collaborative process of identifying and weighing options in order to establish priorities and action [50]. Deliberation is a key component of all the major PSMs [32,40–42].

It is argued that for the planning of sustainable energy projects, risks should be identified and discussed from the start of collaboration in order to answer the question of whether it is beneficial to pursue sustainable energy projects at a local government level. PSMs provide specific theories that can be used as a base from which one can develop a participatory planning approach for sustainable energy at a local government level, but the identification and assessment of risks need to be made explicit. Risk assessment methods on their own focus more on the implementation and project management phases of projects, which are characterised by considerable data and clearly defined boundaries [58] and therefore would not be applicable in the planning phase of local sustainable energy. Marczyk [18] stated that complexity is a fundamental characteristic of every dynamic system found in nature. Marczyk's viewpoints were adapted and therefore the inclusion of risk, even if it is subjective, as part of the approach for local energy sustainability planning is proposed. The identification and assessment of risks should then form part of the deliberation process.

3.5.5. The Development of a Realistic Action Plan Must Be Attainable at the End of a Two-Day Workshop

The requirement of developing a realistic action plan during a two-day workshop is mainly based on practicality and considering the nature of decision making of a local government in a developing country. There are only a few stakeholders (if any) who would be willing to spend more than two days to work on issues that are not part of their day-to-day work. A local municipality is structured for operational management, where decisions are mainly focused on day-to-day operations and limited long-term planning. The time available for long-term planning activities on a municipal level is limited, and therefore careful consideration must be given as to who should be involved, what to discuss and the depth of analysis addressed during these interventions [59]. It is clear from the literature on PSMs and soft OR methods that the aim of all these approaches is to have a better understanding of the complex system, but then to commit to next steps and actions, thereby being action-orientated [60]. Cilliers points out that any plan of action has to be adapted continuously: "If the plan is too rigid—too much central control—the system will not be able to cope with unpredictable changes" [19] (p. 110).

3.5.6. The Approach Must Be Dynamic

When referring to Cilliers's viewpoint [19] that the plan of action needs to be adapted continuously, the approach to be used at a local government level needs to be dynamic. This requirement is in line with PSMs and means that the approach, as well as the associated plans, should be adaptable over time.

3.5.7. The Approach Must Be Formalised with Clear Institutional Arrangements

When drawing on the literature of collaborative governance [13,14,49], maximum effectiveness in the implementation of the EDAS approach can only be achieved if the approach is formalised within a local government; therefore, it should form part of the local government's policies and should include clear roles and responsibilities for all stakeholders, whether private, local, regional or national.

## 4. Introducing Explore, Design and Act for Sustainability

Based on the requirements, as discussed in Section 3.5, a participative planning approach for a sustainable energy future was developed, namely Explore, Design and Act for Sustainability (EDAS). The rationale for this approach is to provide local government with a step-by-step process to facilitate information sharing and discussions on local sustainability, between public and private entities,

in order to determine sustainable energy strategies and to drive energy sustainability. The simple and transparent approach consists of three segments, namely Explore, Design and Act. These three segments form a continuous cycle, based on the philosophies that can be traced back to Galileo Galilei (1564–1642), who believed that conducting designed experiments is the cornerstone of science and the scientific method, and Francis Bacon (1561–1626), who insisted that scientists should proceed through inductive reasoning, from observations to axiom law [61]. These philosophies of Galilei and Bacon led to pragmatism and empiricism [62] and formed the basis of the Japanese plan-do-check-act (PDCA) cycle (1951), as described in Moen [61]. The PDCA cycle and its predecessors consist of steps that are connected in a circle and which represent a "dynamic scientific process of acquiring knowledge" as stated by Stewhart in 1939 [61] (p. 3).

The EDAS approach is therefore built on the Explore, Design, Act (EDA) cycle, as seen in Figure 1. This approach is unique, because it consists of specific parts of current methods and theories, adaptable for application at a local government level. In literature there is a definite move towards mixing methods, because "this allows the field to stay fresh and vibrant as well as allowing the necessary extensions/adaptations to provide the means for managing a broader range of problems–the call for pragmatism" [7] (p. 168).

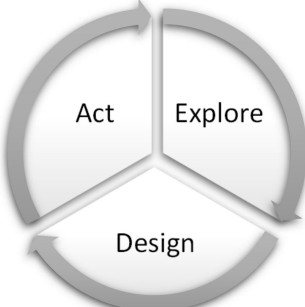

**Figure 1.** The EDA cycle.

### 4.1. Explore to Determine Sustainable Energy Options and Future Conditions

The first segment in the EDAS approach, Explore, aims to determine plausible sustainable energy options within a given context through three steps, namely a) envisage the future, b) determine sustainable energy options and c) identify future conditions. The Explore segment consists of two facilitated workshops and informal interventions over a period as determined by the researchers. The first workshop focuses on public participation and includes, for example, representatives from the communities (wards), business forum representatives, council members and the municipal management team. The second workshop specifically focuses on the decision-makers in the municipality, subject matter experts and identified stakeholders (such as the greatest electricity producers in the area). Informal interventions during the Explore segment are important to understand the municipal context, to build trust and to plan the strategy workshops with the municipal management team.

The aim of the first workshop is to understand the context of the local environment, with the output being the visualised strategy, as shown in Figure 2. In order to develop the visualised strategy in the first workshop, rich pictures and cognitive mapping are used. Rich pictures, a part of SSM, are useful aids to assess complex systems [40]. Cognitive mapping [42] is based on Kelly's personal construct theory [63]. In addition to rich pictures and cognitive mapping, other forms of data collection methods can be used, such as interviews, meetings and observations.

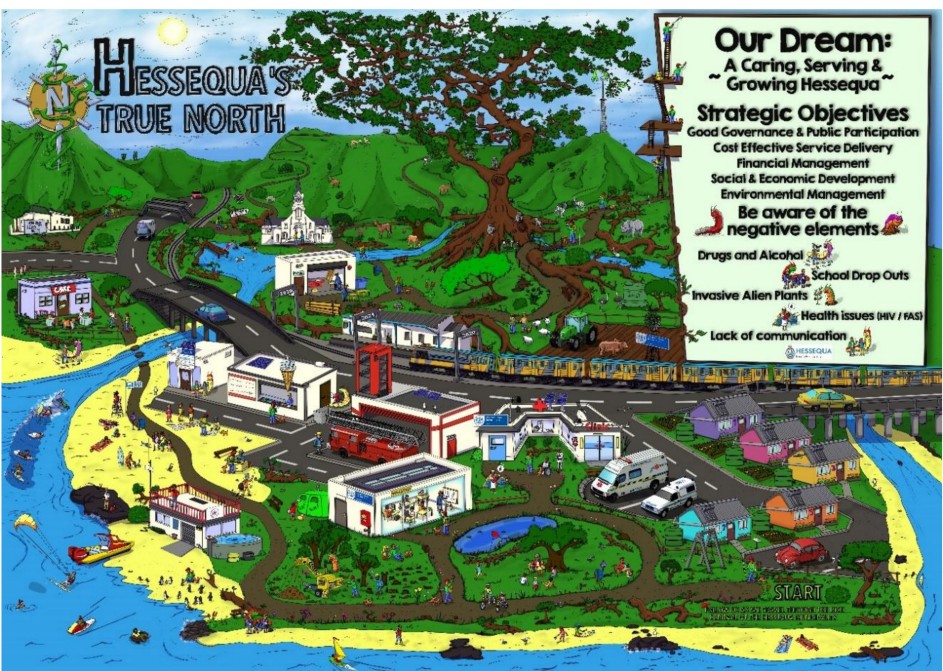

**Figure 2.** Example of a visualised strategy of a local government [25].

The visualised strategy is then used during the second workshop, held over two days, and focusing specifically on a sustainable energy future. The intent of the Explore segment during the second workshop is to creatively think about possible energy futures through firstly, presenting and discussing current trends with regard to sustainable energy and secondly, collecting data on how the participants foresee the specific local area within 20 to 30 years. The written statements about the future indicate the different mindsets and values of the participants. The discussion of these statements helps to gain consensus on where the municipality and its stakeholders see themselves in the future.

After a discussion of the envisaged future, questions are asked to the participants to understand what they perceive as plausible sustainable energy options for the future. This step does not only elicit viable sustainable energy options, based on the knowledge and expertise of subject matter experts, but also eliminates non-contenders early in the planning process. The sustainable energy options determined are taken into the second segment of the approach, namely the Design segment. All the information available on sustainable energy options, such as previous studies done, cost estimates, risks and uncertainties, should be available and discussed when determining the viable sustainable energy options. The data collection of previous information is done during the preparation phase of the workshop. The selection of viable sustainable energy options should be done based on Simon's bounded rationality theory [54], where the aim is to opt for satisfying solutions rather than the optimal solution.

Finally, as part of the Explore segment, potential future conditions are identified. Predictions of future conditions, especially with regard to long-term energy development and planning, are challenging due to the changing environment and many uncertainties. Makridakis, Hogarth and Gaba [64] demonstrated with past examples that accurate forecasting in most areas of business is not possible. Efforts should be channelled to being prepared for different contingencies, rather than to try to predict. McCrone [65] (p. 1) stated that "we can be certain of only one thing–that all predictions about future energy, like all medium-term economic forecasts, will be wrong". Many factors relating to the economy, politics and the environment fall outside of a local government's control, which brings uncertainties. The literature on PSMs describes these factors as uncertainties about related choices beyond the boundaries of the problem field [40], the environment that influences but does not control the system [38] or as a set of 'futures' representative of possible environments of the system [32]. The aim

then with the final step in the Explore segment is to identify, through a subjective process, a set of futures representatives of possible environments of the system that are not within the control of the local government. In order to keep it simple, three possible future conditions are determined, namely (1) a positive outlook, (2) a negative outlook and (3) a most likely outlook. The factors used to determine these future conditions should be agreed upfront with subject matter experts and could include factors as given in the PESTLE analysis (A PESTLE analysis is a framework used by marketers to analyse and monitor the external environment or macro-environmental factors that can impact the operations of an organisation. PESTLE is an acronym for political, economic, socio-cultural, technological, legal and environmental factors) [66].

### 4.2. Design Desirable Sustainable Energy Strategies

The Design segment is built on the foundation of Checkland's soft system methodology, where humans are a part of the system. In a system we see things as being connected, interdependent and working together as a complex whole [40]. The first step in the Design segment is to determine what the system should aim to do. In this regard, Checkland [40] proposed the root definition: a single statement account of the purposeful activity being undertaken by the system. Once the system has been defined, the specific sustainable energy strategies can be developed. A strategy consists of several sustainable energy options implemented within a given timeline. Rosenhead and Mingers [32] referred to configurations and emphasise that special attention should be given when determining the initial commitment. Friend [41] focused on decision areas where all possible options within each decision area are identified. Once these possible options have been determined, the compatible options are grouped together and a list of all the plausible strategies is developed for comparison or evaluation.

The creation of possible strategies from the first principles can result in a list of more than a hundred possible strategies, which is not only difficult to comprehend without computational assistance but will also be very time-consuming when evaluating each of these strategies against the identified futures. Based on the principle of Miller [53], the Design segment then proposes to identify a list of no more than five to nine achievable strategies, keeping the definition of the system in mind. The identified sustainable energy strategies then undergo an evaluation against the possible futures to determine the desirable and undesirable strategies. The deliberation and evaluation are based on the associated perceived risks of each strategy within the identified future. The strategies with the least perceived risks are the most desirable. A discussion of how much risk the decision-makers are willing to take will determine the number of desirable strategies. These desirable strategies will then be used to determine the action steps and way forward to make the desired strategy a reality.

### 4.3. Act for Sustainability

The end state of a PSM is reached when consensus has been reached between the stakeholders on the way forward. In SSM Stage 7 [40], actions to improve the problem situation are discussed and agreed upon. The SCA ends with a commitment package that consists of decisions to be taken now, explorations of the identified uncertainties, deferred decisions and contingency plans [41], and with the RA [32], an agreement on the initial decision is reached. The Act segment then focuses on the development of an action plan, consisting of a description of the specific actions or changes that need to occur, agreement on the champions that will drive the actions and commitment as to when the action steps will be completed.

### 4.4. Evaluation of the EDAS Approach

A two-day Sustainable Energy Journey workshop, to establish a holistic sustainable energy plan for Hessequa Municipality, was held in July 2019 at the municipal offices of Hessequa in Riversdale, Western Cape, South Africa. In order to plan the workshop, the checklist to ensure successful development and implementation of a participatory approach [66] was used. A strong mandate was given by Hessequa Municipality to organise the workshop, participation was free and voluntary,

and careful consideration was given as to which stakeholders, both public agencies and non-state holders, needed to be involved.

The key outcomes of the workshop were to facilitate knowledge sharing, knowledge transfer and networking; to empower the citizens of Hessequa with the opportunity of small-scale embedded generation; to develop different energy strategies for possible energy futures; to identify the obstacles and barriers towards sustainable energy implementation; and to develop a realistic sustainable energy action plan for Hessequa that addresses the development of the energy strategies and plans to remove the obstacles.

The key stakeholders identified to attend the workshop were the municipal management team; the municipal council; Stellenbosch University; the Western Cape government; Eskom Research, Testing and Development; the Centre for Renewable and Sustainable Energy Studies (CRSES) (CRSES at Stellenbosch University facilitates and stimulates research and capacity development activities relating to a vibrant and viable renewable and sustainable energy sector in the southern African region [67]); the Gouritz Cluster Biosphere Reserve (GCBR) (GCBR is a voluntary citizens' initiative dedicated to the conservation of its region's biodiversity, tied to the socio-economic development for the well-being of its peoples. Governed by members, it is a registered non-profit company with the tax status of a public benefit organisation [68]); GreenCape; Garden Route District Municipality; Stilbaai Conservation Trust; and the top 20 electricity users in Hessequa, as well as electrical engineers and energy consultants as subject matter experts. In total, 84 participants were invited to the workshop, of which 28 participants (33%) attended Day 1 of the Hessequa Sustainable Energy Journey workshop, and 23 participants (27%) attended Day 2. The participants were categorised into the following stakeholder groups (with workshop representation shown in Figure 3):

- Western Cape government/GreenCape: grouped together due to their close collaboration with regard to the Energy Security Game Changer (The Energy Security Game Changer aims to ensure sufficient power to sustain households and grow businesses in the Western Cape province, with a goal to achieve an effective 10% contribution to the electricity needs of the Western Cape by 2020 by reducing the province's demand from Eskom [69]), an initiative of the Western Cape government
- The municipal management and council: specifically, Hessequa Municipality
- Stellenbosch University and CRSES
- Municipal stakeholders: people living and/or owning businesses in the Hessequa area (including the top 20 electricity users of Hessequa Municipality)
- Subject matter experts: electrical engineers and consultants
- Other: People not living and/or owning businesses in the Hessequa area and not part of any of the other stakeholder groups.

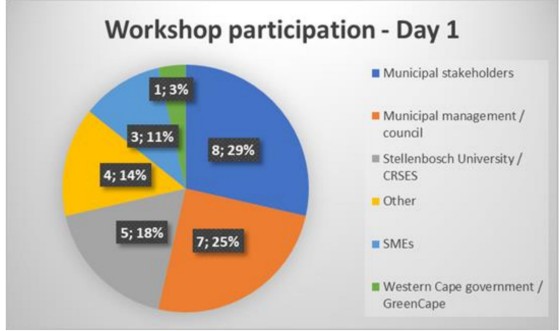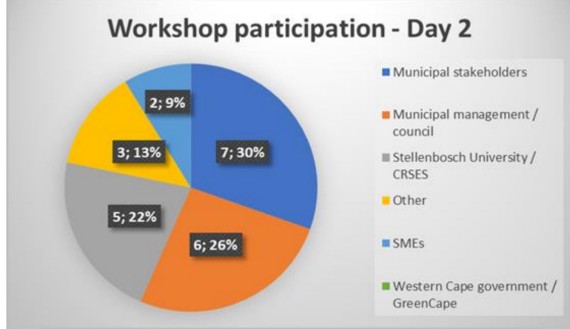

**Figure 3.** Stakeholder representation at Hessequa Sustainable Energy Journey workshop.

The head of the Energy Game Changer of the Western Cape government opened the Hessequa Sustainable Energy Journey workshop with a keynote speech on the energy vision of the Western

Cape. Next, representatives of CRSES gave a presentation on the progress and prospects of renewable energy on a local level. The keynote speech and presentation from CRSES set the scene to kick off the participatory planning approach—EDAS. The EDAS approach structured the discussions and debate to explore and envisage the future of Hessequa; to identify sustainable energy options, uncertainties, obstacles and barriers; to name the Hessequa sustainable energy system; to design sustainable energy strategies and evaluate the strategies against the possible futures; and to lastly develop a sustainable energy action plan for Hessequa.

The main researcher and main author of this paper was in the position to facilitate the EDAS approach at Hessequa, which provided the opportunity to evaluate the approach in terms of practicality and feasibility. The approach starts with divergent thinking, opening the minds of participants to many possibilities without analysing or judging the ideas, followed by a systems perspective to gain an understanding of unintended consequences if projects are implemented in isolation, and ending with convergent thinking, leading to a realistic action plan and way forward. The expectations identified by the participants were met, namely (1) to develop an action plan consisting of simple, adaptable and resilient actions that are realistic in terms of current regulations, (2) to establish ongoing collaboration and (3) to create awareness of energy and environmental sustainability. In addition, six of the seven requirements of the approach, as given in Section 3.5, were met. Although discussions had taken place to formalise the approach within Hessequa, no changes to management procedures or policy amendments were made, to the knowledge of the authors.

Further evaluation was done by asking the participants to complete a workshop evaluation form. The workshop was evaluated not only on the facilitation of the EDAS approach, but also in general. The overall rating of the workshop was based on the amount of new information acquired, expectations met, materials presented, facilitator and presenter skills, and the participatory decision making of Hessequa. In total, 57% of the participants rated the workshop as excellent, 33% as above average and 10% as average. For 90% of the participants, the outcome of the workshop was satisfactory in terms of what was achieved in the available two days. The participants felt that all aspects were covered, and responsibilities were identified and addressed, and that it had been a good start in formulating a sustainable energy plan, with an experience of shared learning and shared visioning with focused discussions. Ten percent of the participants felt that more time was required, especially when keeping in mind that some information was new to many participants. Participants commented that EDAS provided a structured approach that stimulated thinking and facilitated valuable discussions. The approach keeps focus on the subject and cultivates participation towards a desired outcome. A few participants mentioned that the approach needs some refinement and is not necessarily comprehensive enough to reach a detailed plan of action.

In total, 76% of the participants felt that the Explore segment of EDAS was the most valuable, 18% commended the Act segment and 6% viewed the Design segment as the most important. For most participants, exploring the many possible energy options, understanding the advantages and disadvantages of various alternatives and discussing diverse perspectives in understanding the problems were stimulating and valuable. Participants felt that the EDAS approach is a way forward to improve local government participatory decision making. However, care should be taken to ensure that the relevant stakeholders are identified and given an opportunity to participate meaningfully. The role of the facilitator is crucial in such an intervention to ensure that discussions are not side-tracked and that all participants give their input. Other options mentioned to improve local government participatory decision making were better communication and awareness from the municipality, as well as the establishment of community forums, such as an energy forum.

## 5. Discussion

The results from the Hessequa Sustainable Energy Journey workshop show that sustainable energy strategies can be identified in a short period if a diverse group of stakeholders participate and a knowledgeable facilitator ensures that the discussions taking place are structured and focused.

The discussions allowed subject matter experts to share knowledge and information with the other stakeholders, which ensured awareness creation of sustainability issues and their impact on the future environment. The Explore segment followed a divergent approach to consider as many alternatives and options as possible at first, without constraining the participants' thinking. With the aim of later ensuring that the options are realistic, a question was asked to consider the obstacles and barriers for successful implementation. The core barrier perceived, which has an impact on the implementation of sustainable energy options, was the current energy and political landscapes in South Africa, coupled with uncertain and inconsistent regulations. The main obstacle mentioned was the current capital cost (initial investment) of sustainable energy technologies, where return on investment carries the greatest weight in the decision-making process. The current position of the state-owned utility Eskom, with the possibility of future load shedding and aging infrastructure, was not seen as a threat, but rather as an opportunity for local governments to change their energy landscape. What was clear from the discussions is that the reliability of the supply of electricity from Eskom is one of the main factors that will drive decisions and electricity consumers' behaviour.

The Design segment started with explaining to the participants what systems thinking is and then asking them to describe the energy system they would like to design from a systems perspective. Different energy system definitions were identified without a final consensus reached or the selection of a specific system name, but the exercise ensured focus and structure for the next step, namely, to develop energy strategies over a specified timeline. It was made clear to the participants that a strategy can be a combination of different energy options implemented over time. The group discussions on sustainable energy strategies, in the time available, focused mainly on short- to medium-term actions. Participants felt that in order to gain momentum in moving towards a sustainable energy future, it is important to focus on a few selected projects instead of coming up with long-term plans. The proposed actions from all participant groups were then grouped into specific decision areas, which were further used to develop the potential energy strategies. The final step in the Design segment was to evaluate the risk of each of the developed strategies against potential future conditions in order to select a preferred strategy. The identification of potential risks ensured that the different strategies could be deliberated.

The aim with the final segment, namely Act for sustainability, is to agree on an action plan and a way forward. Since the start of the Hessequa Sustainable Energy Journey workshop, it was strongly advocated that a workshop is not worth much if it does not lead to some action. The strategy followed during this segment of the approach was to get verbal commitment on each action item as to who will be responsible for that specific action. The commitment from the municipal director of technical services and the municipal manager was noticeable, and they were comfortable to take on the role of enabling and facilitating the Hessequa Sustainable Energy Journey. The workshop further established willingness from many outside stakeholders (non-state holders) to take part in the journey through contributing and committing to the identified actions. One of the most important next steps agreed during the workshop was to establish an energy forum to continue the discussion and collaboration every quarter, not only on sustainable energy, but also on the sustainability of Hessequa in general.

The authors believe that the EDAS approach can make a difference in moving towards a sustainable energy future in South Africa and other developing countries. More structured discussions taking place between public agencies and non-state holders can ensure a better understanding of the problem of energy sustainability, not only in terms of energy security, but also in terms of climate change. Furthermore, in order to overcome the barriers in terms of legislation, municipalities should continue to exert influence over provincial and national government by taking the lead in enabling and encouraging private sectors to implement sustainable solutions. One of the lessons learnt from conducting the research is that the EDAS approach can only be successful and sustainable if it is formalised and institutionalised as part of government policies. The identification of roles and responsibilities, not only for government bodies, but also for non-state holders, is of utmost importance.

The EDAS approach is a conceptual contribution of new knowledge in the fields of renewable energy and sustainability as well as the fields of soft OR, public administration and policy making.

A practical contribution of facilitating the approach with Hessequa Municipality is given and the ideas that emerged during the workshop can be used by other public agencies to move towards a sustainable future. EDAS provides a practical approach for public participation and collective governance. The novelty of EDAS lies in the flexibility of adapting the methods and tools used, as required by the context and the facilitator. Also, the EDAS approach is not only applicable to sustainable energy and local government, but can be applied to other sustainability challenges, regional and national government, as well as in other organisations.

Future research directions should focus on applying the EDAS approach to more local government contexts, as well as public enterprises. An EDAS facilitation guide is included as Supplementary Materials for this purpose. Another research focus is to determine how risk factors are being identified in the decision making of sustainable solutions and how these risk factors are used in the deliberation process to influence the decisions, especially in terms of prioritising the different strategies. In terms of moving towards a sustainable energy future, further research could focus on the human behaviour elements with regard to sustainable energy decisions, especially on how mindsets can be influenced to focus more on longer-term environmental benefits than on short-term financial gains.

## 6. Conclusions

The main aim of the research on which this paper reports was to develop a participatory planning approach for local governments in South Africa that can be used during participation workshops concerning long-term sustainability issues such as energy. Due to the complex nature of local energy sustainability, it has been determined that the participatory planning approach must incorporate the characteristics of a complex problem. Complex problems consist of a system of problems with multiple stakeholders and perspectives. The complex structure of these problems, with many interrelated issues, tends to lead to unintended consequences, where a solution to one part of the problem could influence other parts of the system. This also means that complex problems are characterised by uncertainty and risk. The literature study concluded that complex problems can never be understood or solved in full, therefore great care should be taken when addressing complex problems.

Building on recognised methodologies and approaches in the field of soft OR, while taking into consideration the context of local governments in South Africa, and using public participation and collaborative governance as an overarching framework, a new approach, EDAS, was developed to explore, design and act for sustainability. The EDAS approach was successfully applied and evaluated in a workshop with Hessequa Municipality, a local government in the Western Cape Province of South Africa, to determine sustainable energy strategies and a way forward. The developed approach is novel due to its holistic, dynamic and transparent features, and it provides a structure for focused discussions between the public sector and local, regional and national government.

A limitation of the research is that the EDAS approach was only applied and evaluated in one local government context in South Africa. It is recommended that more research be conducted on applying and evaluating the EDAS approach in other local government contexts as well as in public enterprises. More cycles of the approach need to be applied to improve and refine the approach in terms of developing more detailed sustainable energy strategies, and to ensure that the approach can be formalised and institutionalised as part of a local government.

**Supplementary Materials:** EDAS facilitation guide is available online at http://www.mdpi.com/2071-1050/12/3/862/s1: (see pdf attached).

**Author Contributions:** Conceptualisation, E.F.; Formal analysis, E.F.; Methodology, E.F.; Supervision, A.B.; Validation, E.F.; Writing—original draft, E.F.; Writing—review and editing, A.B. All authors have read and agreed to the published version of the manuscript.

**Funding:** This research received no external funding.

**Acknowledgments:** The authors would like to thank Hessequa Municipality, Western Cape, South Africa for taking part in the research and for their support and assistance throughout the process.

**Conflicts of Interest:** The authors declare no conflict of interest.

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
