# Peer review of "Explore, Design and Act for Sustainability: A Participatory Planning Approach for Local Energy Sustainability"

_sustainability, doi:10.3390/su12030862_

Round 1

Reviewer 1 Report

This is an interesting paper that well presents the application of the authors' suggested "EDAS" approach--as a participatory planning process--to Hessequa's two-day sustainable energy workshop. The draft can be improved to inform the lessons better, policy implications, or new knowledge generated by and learned from the authors' actual trial.

There is a weak linkage between the reviewed literature and the development of EDAS and its application. Current literature focused on the overview of complex problems and soft OR/PSM. Participatory planning is not only approached from the viewpoint of the combination of soft and hard systems methods but also more significantly taken by new public service/open government innovation approach as well as collaborative governance. I would recommend shortening existing literature reviews on complex problem and PSMs, rather include upper mentioned approaches.

The development process and rationale for "EDAS" approach--which is the main idea and purpose of this study--is missing. Any references, rationalization needs to be addressed.

Please give more detailed information on the methods and results/implications of the application of each stage of your EDAS model, so that readers--either academics or practitioners--could learn from this paper.

Author Response

Dear Reviewer

Thank you for your review and the valuable input. 

I have updated the manuscript based on the comments. I have indicated these changes with comments in the manuscript. Also see detail feedback on each reviewer comment. 

The manuscript was also language edited and all references have been checked.

Kind regards

Elaine

Reviewer 2 Report

In this paper, the authors are focusing on the development of a participatory planning approach for local

energy sustainability. A novel participatory approach was then applied, verified and validated in a facilitated workshop with a local municipality in the Western Cape province of South Africa.

In general, the topic of this paper is interesting. The subject could be relevant and appropriate for the Sustainability. Here follow some points that need further attention:

- Introduction and background sections: Please, be more critical in addressing the research gap. What is the contribution of the paper to the literature? Emphasize these aspects already in the introduction to make paper attractive for readers.

- A Discussion section is necessary. Authors should discuss the results and how they can be interpreted in perspective of previous studies and of the working hypotheses. The findings and their implications should be discussed in the broadest context possible. Future research directions may also be highlighted.

- Limitations of the study must be added in the Conclusions section.

- The Figures do not have references. The authors must add the source to each Figure.

- The reference list must be reviewed. Please read the Author’s guidelines and Sustainability template and make the changes accordingly.  

- 84 reference is too much for a research paper. The authors should make the reference list shorter by keeping only the relevant references.

Author Response

(The authors gave the same response as above.)

Round 2

Reviewer 1 Report

Thank authors for your effor to improve the draft according to the comments recommended.